# Horse-Riding Competitions Pre and Post COVID-19: Effect of Anxiety, sRPE and HR on Performance in Eventing

**DOI:** 10.3390/ijerph17228648

**Published:** 2020-11-21

**Authors:** Sabrina Demarie, Christel Galvani, Veronique Louise Billat

**Affiliations:** 1Department of Movement, Human and Health Sciences, University of Rome “Foro Italico”, Piazza de Bosis 6, 00135 Roma, Italy; 2Applied Exercise Physiology Laboratory, Department of Psychology, Università Cattolica del Sacro Cuore, Largo A. Gemelli, 1, 20123 Milano, Italy; christel.galvani@unicatt.it; 3Unit of Integrative Biology of Adaptations to Exercise, EA 7362, Université d’Evry-Val d’Essonne, Genopole, 91037 Evry, France; veroniquelouisebillat@gmail.com

**Keywords:** COVID-19, state anxiety, rate perceived exertion, horse-riding, competition

## Abstract

The aim of the present study was to quantify the impact of training restrictions, due to COVID-19 sanitary emergency, on physical and emotional strain of horse-riding Eventing competitions before and after eight weeks of lockdown. Performance was assessed by the penalty points attained, anxiety by the Competitive State Anxiety Inventory-2, strain by the Rating of Perceived Exertion (RPE) method. Moreover, Heart Rate was continuously monitored for fifty-four female national level Eventing horse-riders. Lockdown decreased performance outcome of horse-riders in Eventing competitions up to six weeks, with the Dressage test being the most affected discipline. Performance in Dressage was strongly related to both anxiety and session-RPE. After lockdown, Show-Jumping and Cross-Country courses were shorter allowing RPE to remain stable, session-RPE to significantly decline and cardiovascular strain not to exceed pre-lockdown values. In conclusion, emotional stress in Dressage and workload in Cross-Country should be carefully managed by equestrian Eventing stakeholders when planning training and competitions after a period of lockdown. Moreover, sRPE appears to offer a practical method of monitoring riders load during training and competition and could also be of use for home-based training during any future sport activities restrictions.

## 1. Introduction

To decrease the pathogen spread of SARS-CoV-2, extraordinary measures were implemented by the Italian government from 8 March to 4 May 2020 limiting physical activity and sport. Outdoor jogging and running were permitted only in a 200 m range from home, and any other physical activity and competition was prohibited. After eight weeks of lockdown elite athletes undertaking individual outdoor sports were entitled to resume training, whilst closed-door competition recommenced six weeks later.

Athletes are aware of detrimental effects of physical activity restrictions, and the majority of them have attempted to train at home within the constraints of the equipment and space that they had available. Nonetheless, training restrictions have led to concerns about athletes’ ability to maintain key physical and mental attributes consequently having the potential to impact on performance and injury risk on resumption of training and competition. Therefore, greater attention than usual should be payed to workload, perception of exertion and signs and symptoms of injury [1]. As is well known, the initial weeks of training should focus on the adjustment to the intensity and duration required for training and competition, which should be progressed gradually. In general, a period of around six weeks of preparation is likely to be sufficient for most athletes to return to being competition ready [2].

However, equestrian disciplines have different demands compared to many other sports, since the combined horse–human partnership adds further complexity. The human athletes not riding a horse can only train their physiological traits, technical skills and the crucial relationship with the horse will be impaired. Likewise, the equine athlete, to be technically and mentally prepared for competition, needs daily practice with its human partner in the field. During the eight weeks of lockdown, most horses could only undergo thirty to forty minutes a day of nonmounted round pen exercises at the three gates. Even worse, they were led by a trainer that usually was not their owner or their usual rider. It can be easily presumed that an insufficient or inappropriate horse training and a premature participation in competition could lead to reduced standards in horse’s welfare. Indeed, the role of psychological components in equestrian performance has been studied and the influence of intensity of precompetitive somatic arousal on equestrian performance has been acknowledged, suggesting that is vitally important that equestrian athletes learn to be in control of both their own body and mind [3,4,5]. Therefore, both physical and emotional strain should be carefully monitored and dealt with when managing workload in horse-riding.

Information about the physiological and emotional demands of horse-riding can be collected by making relevant observations during competition. Horse-riding Eventing competition comprising Dressage, Show-Jumping and Cross-Country tests is described by the Fédération Equestre Internationale (FEI) as “the most complete combined equestrian competition” [6]. Due to the complexity and completeness of Eventing, understanding the impact of training reduction or even cessation on subsequent workload administrations is critically important. Due to the unique circumstances created by COVID-19 induced lockdown, the need to provide athletes and coaches with reliable evidence-based resources regarding return to training and competitions is even more necessary.

A study aimed to estimate the psychological impact of COVID-19 and related restrictive measures on the Italian general population during lockdown reported that 52.4% of participants experienced poor sleep and anxiety. Authors also expected mental health symptoms to persist beyond this critical situation [7]. It is perhaps not surprising then, that acute stress disorders, anxiety, irritability, poor concentration and indecisiveness, deteriorating work performance, post-traumatic stress disorders, high psychological distress, depressive symptoms and insomnia are reported as consequences of quarantine [8]. Bompa and Buzzichelli [9] suggest that an abrupt cessation of training by highly trained athletes creates a phenomenon known as detraining syndrome, characterized by insomnia, anxiety, depression, alterations to cardiovascular function and loss of appetite. Mental fatigue has been demonstrated to negatively influence performance [10] possibly due to an alteration in the perception of effort when individuals are mentally tired [11]. Mental fatigue represents sensations of tiredness experienced during or after prolonged periods of cognitive activity that often influence decision making, attention, motivation, and the voluntary willingness to resist fatigue. Therefore, to ensure that athletes adhere to the program, training load should be individualised and adapted to the possible sensation of tiredness induced by the lockdown, modifying the ratio between prefrontal activation and Rating of Perceived Exertion (RPE) [12]. Consistently, recent studies [13] have suggested that the brain regulates exercise performance based on afferent information from peripheral physiologic receptors that give rise to sensations (e.g., nausea and thirst among others). Indeed, not only physiological, and neural determinants explain the variation of RPE but also personality factors (extraversion, neuroticism, depression and anxiety) were shown to affect RPE [14]. Therefore, application of the RPE method can help to understand the multifaced influence of lockdown on athlete’s fatigue and stress and to carefully manage the athletes back to full training and competition. Therefore, the aim of the current study was twofold: firstly, to quantify the impact of horse-riders training restrictions on physical and emotional strain of Eventing competitions after eight weeks of lockdown due to the COVID-19 emergency; and second, to provide a scientific evidence base for the provision of sport-specific strategy guidelines for Eventing training and competition.

## 2. Materials and Methods

Ninety-eight male and female national level Eventing horse-riders, participating at national and international competitions, were recruited from eight clubs, with forty-four athletes being excluded from the study. Subjects were excluded if: they declined to participate (n = 6), had musculoskeletal injury within 6 months of the start of the study (n = 2), had pulmonary distress symptoms within 6 months of the start of the study (0), had musculoskeletal pain at the time of testing (>0 on a 0–10 numeric pain scale) (n = 1) or were eliminated during competitions (n = 26). Of the sixty-three remaining subjects fifty-four were females; to eliminate gender bias, male data were excluded from the study (n = 9).

As a result, fifty-four female Eventing horse-riders participated in the study. Their age, height, and body mass mean (SD) were 22.1 (1.9) years, 164.1 (8.2) cm and 55.8 (4.1) kg. Participants had been competing in Eventing for 78 (5.30) months at least. During the 3 months before the study they had completed 15.2 (2.1) training sessions each week on 6.1 (0.4) days per week. The average total training time each week was 12.3 (2.8) h, comprising 4.9 (0.9) h dressage, 4.3 (1.1) h show-jumping and 3.1 (0.7) h horse cross-country. During lockdown, athletes trained on their own on a regular basis for 1.5 h 5 days/week, routine comprised jogging and stretching warm-up, light weights lifting (0.5 kg and 1 kg) for the upper and lower limbs and no-load core exercises.

A within-subject design was used evaluating two competitions held 4 weeks before lockdown (pre) and two competitions 10 weeks after training had recommenced (post). For each competition, DressagePre and DressagePost, Show-jumpingPre and Show-jumpingPost, Cross-countryPre and Cross-countryPost, state anxiety, heart rate (bpm) and RPE were assessed.

Following approval of the project by the institutional Research Ethics Review Committee (University of Rome Foro Italico, Rome, Italy, Commissione di Ateneo per la Ricerca, CAR 53/2020, 21 June 2020), and after receiving permission from the clubs, all riders were informed of the general purpose of the study, of their rights as study participants and of the anonymity of their data and provided written informed consent.

Performance in competition is reported as penalty points (PPoints) valuated by official jury. For the final classification, the winner is the athlete with the lowest total PPoints. Elimination from one test entails results in immediate elimination from the competition.

One hour before each test, subjects completed the Italian version of the Competitive State Anxiety Inventory-2 (CSAI-2) by Martens et al. [15]. The CSAI-2 was used to measure cognitive anxiety, somatic anxiety, and self-confidence among the participants. It has 27 items with nine items in each of the three subscales. Possible scores for each subscale ranged from 9 to 36. Respondents rate on a 4-point scale the extent of their agreement using anchors of (1) Not at all and (4) Very much. Higher scores on the subscales somatic and cognitive anxiety reflect higher levels of anxiety and a higher score on the self-confidence subscale indicates higher levels of self-confidence. The CSAI-2 measures temporal states, rather than stable personality traits and has good relationship with equestrian performance. In general, anxiety is made up of cognitive and somatic components. Cognitive anxiety encompasses worries or concerns about potential failure and the adequacy of one’s performance as well as disrupted attention and negative expectations. The somatic aspects of anxiety are seen as comprising the autonomic arousal with its physiological responses such as sweating and increased heart rate. Cognitive anxiety is by definition indicative of negative expectations and focused on negative thoughts in the form of worry. Anxiety, as an emotional response, can significantly influence successful return to sport. In addition, anxieties related to the inability and/or uncertainty to return to previous level of performance and lack of athletic appearance have been found to influence a successful return to sport process. Moreover, lack of athletic identity, feelings of isolation and pressures to return to sport when the athletes themselves do not feel ready to return are also typical emotional responses during the return to sport phase, and they are likely to increase anxiety if not addressed. Moreover, athletes’ affective responses are proven to be related to perception of exercise intensity, being that higher intensities were associated with less favourable affective responses. On the day before the competition participants were provided with the Italian version of the CR-10 scale [16], which they were asked to familiarize themselves with. Half an hour after the race participants were asked to give a nominal score to describe personal RPE of “mean training intensity” during that competition test. Afterwards, the session-RPE (sRPE) method was applied to assess exertion during Eventing competition [17].

After warm-up, heart rate (HR) was continuously monitored during each competition (HRM-Dual™ Ant+, Garmin, Olathe, KS, USA). Data recorded between the start and end of each test were assessed for peak, minimum and average HR. The peak, minimum and average HR percentage of the maximal theoretical heart rate (%max) were also estimated according to the procedures used by Tanaka et al. [18].

### Statistical Analysis

Data were expressed as mean ± SD. Normality of the distribution of the orientation errors was checked with the Shapiro–Wilk test. In case of skewed distribution, differences between the pre and post lockdown were tested using Mann–Whitney U. In case of normally distributed data, differences pre and post were tested using a paired *t*-test. Differences between Dressage, Show-jumping and Cross-country were tested using a Kruskal–Wallis test. In case of normally distributed data, differences were tested using a One-Way ANOVA. Post hoc analysis was performed using the Bonferroni post-hoc test as appropriate. To understand possible relationships between performance, anxiety, sRPE and HR, Spearman Rank correlation or Pearson correlation analyses were applied when proper. Significance was accepted at the level of *p* < 0.05.

## 3. Results

No significant differences in PPoints for Show-jumping and Cross-country were reported between pre and post lockdown. On the contrary, a significantly worse performance (higher total PPoints) post lockdown was observed for the Dressage test led by the significantly higher total PPoints. Similarly, significantly higher post lockdown values were found for anxiety of the Dressage test only, while Show-jumping and Cross-country anxiety did not significantly change significantly. RPE was significantly higher in the post lockdown Dressage test but remained unchanged (pre and post COVID-19) in the Show-jumping and Cross-country tests. When duration of the tests was taken into account, sRPE was still significantly higher for the post lockdown Dressage test but significantly lower for the post Show-jumping and Cross-country tests. On the other hand, test duration (time) stayed constant in the Dressage tests but significantly was shorter in the Show-jumping and Cross-country tests (Table 1).

After lockdown HRmin increased significantly both in absolute and relative to maximal values in the Dressage test only; HRavg increased in the Dressage and Show-jumping tests and HRpeak remained constant in the three tests. As percentage HRmax, HRpeak always reached <100% both in the pre and post-tests (Table 2).

For both the pre and the post values, all differences were statistically significant among the three tests. Post hoc analysis revealed that PPoints were always significantly higher in Dressage than in Show-jumping and Cross-country. No differences were found between Show-jumping and Cross-country for PPoints, anxiety and HRpeak, whereas Dressage showed statistically higher anxiety and lower HRpeak values than Show-jumping and Cross-country. No differences were found between Dressage and Show-jumping for HRavg and HRpeak in the post values only (Table 3).

For the relationship between measured parameters, both in the pre and post Dressage tests PPoint resulted significantly related to sRPE (pre: r = 0.474, *p* = 0.0002; post: r = 0.457, *p* = 0.0004) that, in turn, was significantly related to anxiety (pre: r = 0.458, *p* = 0.0004; post: r = 0.433, *p* = 0.0009). A weak but significant correlation was found between anxiety in Dressage and PPoints in the post test (r = 0.322, *p* = 0.0174), while in the pretest, correlation did not reach significance (r = 0.253, *p* = 0.0648).

## 4. Discussion

Results of the present study found that eight weeks of training restrictions and competition avoidance, due to COVID-19 necessitated lockdown, decreased performance of horse-riders in Eventing competitions. The present study also demonstrated that, among the three Eventing tests, Dressage test resulted the Eventing discipline most affected from lockdown. Moreover, results are likely to substantiate the contribution of cognitive distress to overall perception of effort and to performance outcome in Dressage.

The higher PPoints and the greater physiological demands indicate a decreased performance. Even though HRavg remained in a moderate to vigorous intensity domain [19] ranging between 60 and 89 percent of the maximal theoretical heart rate in all tests, HRpeak values always resulted <100%, both for the pre and post lockdown competitions. Previous studies on the acute physiological responses to a long duration show jumping riding performance in the obstacle test track in female show-jumping and Eventing riders, reported that participant reached almost 100% of their heart rate already at the beginning of the show jumping test track. Authors speculated that a static riding posture, long-term maintenance of the body control and stress during the obstacle track may increase HR more than one would expect. Indeed, during jumping 94% of the maximal HR has been obtained at only 75% of maximal oxygen uptake [20]. Hence, the demands placed on the cardiovascular system emphasize that horse riders are required to perform at an ever-higher percentage of their maximal aerobic capacity throughout the three Eventing tests. As stated by the fundamental principles of training periodization, after a break, training sessions should focus on the refamiliarization with the qualities required for competition, with intensity and duration progressing gradually. Accordingly, the shorter competition format performed after lockdown enabled athletes to maintain a tolerable physiological demand, preventing an excessive cardiovascular stress through the one-day Eventing. This is also in compliance with the Italian Federation of Sport Medicine precaution guidelines [21] for a safe return to training and competition after lockdown [22,23].

Between the three Eventing discipline, Dressage test resulted the most affected from lockdown. In fact, anxiety and RPE resulted significantly increased post lockdown in the Dressage test only; and unexpectedly they remained unchanged in the Show-jumping and Cross-country tests. The latter results could be explained by the significantly shorter duration of competition in the post Show-jumping and Cross-country tests, while competition duration was unmodified in the Dressage post-test. In this regard, it has to be pointed out that after lockdown most athletes chose to compete in the Short Format Competition, where the level of difficulty is similar to the Long Format according to the star system, but the Cross-country course was 40% shorter and the Show-jumping courses were kept as short as possible within the rule’s framework (15% shorter). The shortened duration of Show-jumping and Cross-country in the post tests could explain the stable RPE values and the significantly lower sRPE in the post Show-jumping and Cross-country tests.

Performance in Dressage was strongly related to both anxiety and sRPE, corroborating the contribution of cognitive distress to perception of effort and to performance outcome. At this regard, Jerome and Williams [24] suggested that in sports with high memory demand, the increased worrying associated with higher levels of cognitive anxiety could put a drain on athletes’ resources and limit their attentional capacity. Accordingly, in the discipline of dressage, greater levels of perceived intensity of precompetitive arousal was reported to have a debilitating effect on dressage performance as well as horse-rider interaction scores, while in Show-jumping no such relationship was found [5]. Authors claimed that in dressage, the riders must memorize a complex pattern of movements, while Show-jumping courses are built in a fairly logical progression. Therefore, the greater demand on memory capacity in the sport of dressage may lead to a negative relationship between precompetitive arousal and performance. Moreover, being competitive Dressage performance based on judge’s aesthetical and technical evaluation only, riders may be more susceptible to high levels of state anxiety, while Show-jumping and Cross-country PPoints derive from the mathematical calculation of faults and time. Since Dressage constitutes the basis for many equestrian disciplines, including jumping and Eventing [25], it can be presumed that the ability to communicate effectively with the equine partner becomes a fundamental skill that all equestrian athletes must master [4,5].

Horse riding relies on subtle interactions between the horse and the athlete, performance not depending on human ability and skill only but equally on that of the equine partner and most importantly on the quality of the interaction between the horse and the rider [4,25].

To be able to perform “as one”, the aids given by riders should be as subtle and refined as possible, demanding considerable levels of fine motor control and accuracy. Hence, riders’ low levels of arousal are probably advantageous. Increases in muscular tension, respiratory rate, and heart rate are all likely to affect the rider’s ability: first, to transmit an aid, and second, to release the aid again as quickly as possible afterward. Equestrian sports could be anxiety-inducing, especially in times of increased mental and physiological stress such as the return to competitions after a forced interruption of training. Riders may tend to “hold on” too tightly to the reins, or block the horse with their seat, as higher muscular tension would prevent them from relaxing the relevant muscle groups [26]. Physical tension resulting from increased levels of anxiety is detected by the horse, who, in turn, reacts with physiological symptoms associated with anxiety and fear. Any signal indicating anxiety from the rider, such as increased muscle tonus, respiratory or heart rates, may alert the horse, making it fear imminent danger, which is likely to make it react with a flight response, usually leading to faults or loss of marks [4].

Riders mental skills training aimed at improving emotional composure and physical relaxation is likely to also have a beneficial effect on the horse. A study conducted on horses participating in elite show jumping and dressage competitions reported a rather high incidence of conflict behaviour indicative of discomfort, confusion, and resistance or hyperactivity. Conflict behaviours increased in association with increasing levels of complication or difficulty of the tasks encountered, maybe indicative of some shortcoming in the preparation of the equine athletes including overtasking or some other measures of internal conflict or stress being experienced by the horse while being ridden in competitions [27]. Competing with physically or psychologically unprepared horses could lead to mental and/or physical damage to the animal, therefore return to competitions should be carefully planned to ensure not only successful performances but mostly horses’ welfare.

Social isolation, exercise reduction, sedentary behaviour and changes in nutrition have a psychological consequence and can possibly impact sleep and fatigue of athletes [28]. Indeed, it can be assumed that taking part in a competition with merely six weeks of training, after almost two months of lockdown, can cause a notable amount of emotional distress. Therefore, in horse-riders it is likely that, alongside the physical health of the athlete, their mental well-being is affected by sport activity limitations, highlighting the need for well-defined and accessible support structures for athletes and staff, both during and after isolation [2].

This study reported some particular aspects, strengths and practical applications. It is the first to quantify the impact of horse-riders training restrictions on physical and emotional strain of Eventing competitions after eight weeks of lockdown due to the COVID-19 emergency. One strength is the use of objective measures of sports performance, anxiety, exertion, and effort. Moreover, RPE appears to offer a practical method of monitoring athletes’ workload during training and competition, but it could also be of use for home-confined training during any future sport activities’ restrictions. It is important to mention that RPE does not concern athletes only but can also provide a needed guidance for the general population fitness maintenance. Lastly, emotional stress in Dressage and workload dependent from course length in Cross-Country should be carefully managed by equestrian Eventing stakeholders supervising athletes, during this unprecedented period of restriction and when planning training and competitions after a period of lockdown.

Several limitations need to be acknowledged. First, the present study is limited to young female high-level athletes, therefore results may not be applicable to male or beginner horse-riders. Nonetheless, the use of data from studies on elite level competitions will allow novice rider’s coaches to imitate high level riders for appropriate training prescription and profitable competition strategy. Second, the data are correlational, thus conclusions concerning causal relations cannot be drawn. Third, the scales used to measure somatic and cognitive aspects and self-confidence of sport competition trait anxiety were not intended to differentiate between these three aspects of anxiety.

Future research is required to better understand the impact of several weeks of training restrictions, due to any emergency state, on physical and emotional strain of horse-riding Eventing competitions in different gender population and competition levels.

## 5. Conclusions

Eight weeks of training restrictions and competition avoidance due to the COVID-19 sanitary emergency decreased the performance of horse-riders in subsequent Eventing competitions held six weeks after training resumption, with the Dressage test being the most affected discipline. Moreover, performance in Dressage was strongly related to both anxiety and sRPE, corroborating the contribution of cognitive distress to perception of effort and to performance outcome in such an emotionally driven discipline. Finally, in compliance with the guidelines for return to training and competition after the COVID-19 emergency, the shorter Show-jumping and Cross-country courses in the post lockdown Eventing competitions allowed RPE to remain stable, sRPE to significantly reduce and cardiovascular strain to remain stable at pre-lockdown values for all three types of test.

## Figures and Tables

**Table 1 ijerph-17-08648-t001:** For Dressage (DR), Show-jumping (SJ) and Cross-country (XC) competition performance (PPoints), anxiety, rating of perceived exertion (RPE), test duration (Time) and session RPE (sRPE) values are reported as mean ± standard deviation, *p*-values are reported for differences between pre and post lockdown competitions.

	Pre	Post	*p*-Value
**PPoints**			
DR	33.1 ± 3.4	57.7 ± 6.7 *	<0.0001
SJ	3.4 ± 4.1	4.4 ± 3.8	0.2203
XC	2.5 ± 3.8	3.9 ± 5.2	0.1736
TOT	39.0 ± 6.5	66.0 ± 7.8 *	<0.0001
**Anxiety**			
DR	41.3 ± 8.2	43.7 ± 7.4 *	<0.0001
SJ	36.6 ± 6.8	36.9 ± 7.7	0.1676
XC	37.2 ± 8.6	37.9 ± 8.5	0.0592
**RPE**			
DR	5.1 ± 0.6	6.6 ± 0.5 *	<0.0001
SJ	6.8 ± 0.6	6.9 ± 0.9	0.3482
XC	8.9 ± 0.6	8.9 ± 0.5	0.8590
**Time (s)**			
DR	330.9 ± 16.0	330.3 ± 16.4	0.7730
SJ	78.4 ± 2.3	67.0 ± 2.9 *	<0.0001
XC	258.7 ± 9.2	156.0 ± 21.0 *	<0.0001
**sRPE**			
DR	1678.2 ± 236.9	2166.6 ± 218.8 *	<0.0001
SJ	526.1 ± 40.8	461.0 ± 47.4 *	<0.0001
XC	2353.3 ± 167.4	1377.6 ± 176.5 *	<0.0001

* Indicates *p* < 0.05.

**Table 2 ijerph-17-08648-t002:** For Dressage (DR), Show-jumping (SJ) and Cross-country (XC) competition minimum (HRmin), average (HRavg), peak (HRpeak) heart rate absolute and relative to (%max) values are reported as mean ± standard deviation, *p*-values are reported for differences between pre and post lockdown competitions.

	Pre	Post	*p*-Value
**HRmin (b** **·** **min** **^−1^** **)**			
DR	87.5 ± 9.3	112.0 ± 7.5 *	<0.0001
SJ	118.9 ± 8.4	120.1 ± 10.2	0.5042
XC	81.7 ± 9.9	81.6 ± 7.6	0.9355
**HRavg (b** **·** **min** **^−1^** **)**			
DR	145.7 ± 11.0	151.0 ± 10.3 *	0.0109
SJ	147.6 ± 8.8	152.2 ± 9.9 *	0.0116
XC	133.7 ± 9.2	136.3 ± 8.6	0.1079
**HRpeak (b** **·** **min** **^−1^** **)**			
DR	197.6 ± 3.6	198.3 ± 3.3	0.1814
SJ	200.0 ± 4.0	199.6 ± 2.4	0.3036
XC	199.9 ± 6.0	200.6 ± 5.6	0.4894
**%HRmin (%max)**			
DR	45.5 ± 4.9	58.2 ± 4.0 *	<0.0001
SJ	61.7 ± 4.5	62.4 ± 5.4	0.5078
XC	42.4 ± 5.2	42.4 ± 4.0	0.9325
**%HRavg (%max)**			
DR	75.7 ± 6.0	78.4 ± 5.4 *	0.0112
SJ	76.7 ± 4.6	79.1 ± 5.3 *	0.0114
XC	69.5 ± 4.9	70.8 ± 4.6	0.1070
**%HRpeak (%max)**			
DR	102.7 ± 2.1	103.0 ± 1.8	0.1862
SJ	103.9 ± 2.0	103.7 ± 1.4	0.3095
XC	103.8 ± 3.1	104.2 ± 3.1	0.4851

* Indicates *p* < 0.05.

**Table 3 ijerph-17-08648-t003:** Differences among dressage (DR), show-jumping (SJ) and cross-country (XC) for competition performance (PPoints), anxiety, rating of perceived exertion (RPE), test duration (time), session RPE (sRPE) and minimum (HRmin), average (HRavg), peak (HRpeak) heart rate absolute and relative values are reported for pre and post lockdown competitions.

	Pre-Lockdown			Post-Lockdown		
	ANOVA or Kruskall–Wallis (*p*) Bonferroni or Mann–Whitney (*)					

	DRvsSJ	DRvsXC	SJvsXC	DRvsSJ	DRvsXC	SJvsXC
PPoints	<0.0001			<0.0001		
	*	*	ns	*	*	ns
Anxiety	0.0042			<0.0001		
	*	*	ns	*	*	ns
RPE	<0.0001			<0.0001		
	*	*	*	*	*	*
Time (s)	<0.0001			<0.0001		
	*	*	*	*	*	*
sRPE	<0.0001			<0.0001		
	*	*	*	*	*	*
HRmin (b·min^−1^)	<0.0001			<0.0001		
	*	*	*	*	*	*
HRavg (b·min^−1^)	<0.0001			<0.0001		
	ns	*	*	ns	*	*
HRpeak (b·min^−1^)	0.0156			0.0119		
	*	*	ns	ns	*	ns
HRmin (%HRmax)	<0.0001			<0.0001		
	*	*	*	*	*	*
HRavg (%HRmax)	<0.0001			<0.0001		
	ns	*	*	ns	*	*
HRpeak (%HRmax)	0.0181			0.0171		
	*	ns	ns	ns	*	ns

* Indicates *p* < 0.05.

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
