# Peer review of "Horse-Riding Competitions Pre and Post COVID-19: Effect of Anxiety, sRPE and HR on Performance in Eventing"

_ijerph, 2020, doi:10.3390/ijerph17228648_

Round 1
Reviewer 1 Report
This is a useful paper and produced in a timely manner given the time line of COVID19 related restrictions and lockdown. The paper is interesting and the findings of value to practitioners. The text needs some further revision to address a few language and wording shortcomings. I have provided a series of comments on the submitted manuscript. I hope that you find them useful.

Author Response
First of all, I am really glad you appreciated our work, I found your suggestions very usefull, and I hope I succeed in fulfill all of them.
Commented [A1]: Female riders only?
At the beginning males were involved, but excluded from subjects group to eliminate gender bias as explained at lines 98-99
Commented [A2]: Sentence
The aim of the present study was to quantify the impact of training restrictions, due to COVID-19 sanitary emergency, on physical and emotional strain of horse-back Eventing competitions before and after 8 weeks of lockdown
Commented [A3]: Make a separate sentence.
Performance was assessed by the penalty points attained, anxiety by the Competitive State Anxiety Inventory-2, strain by the Rating of Perceived Exertion (RPE) method. Moreover, Heart Rate was continuously monitored for fifty-four female national level Eventing horse-back riders.
Commented [A4]: In what sense? Commented [A5]: Rephrase
In conclusion, emotional stress in Dressage and workload in Cross-Country should be carefully managed by equestrian Eventing stakeholders when planning training and competitions after a period of lockdown.
Commented [A6]: Reword?
the initial weeks of training should focus on the adjustment to the intensity and duration required for training and competition
Commented [A7]: In Italy?
in Italy and in southern France
Commented [A8]: On n horses?
horse cross-country
Commented [A9]: Is this HRmean?
yes, it is HR mean, we named it average only to clearly distinguish it from HRmin (minimum), if you prefer we will change average with mean
Commented [A10]: Rephrase – for relationships between ……
For relationship between measured parameters, both in the pre and post DR tests PPoint resulted significantly related to sRPE
Commented [A11]: On what basis?
because Hyttinen et al., 2020, during jumping 94% of the maximal HR has been obtained at only 75% of maximal oxygen uptake (high %HRmax in face of a medium %VO2max)
Commented [A12]: Split this into 2 or 3 shorter sentences.
As stated by the fundamental principles of training periodization, after a break, training sessions should focus on the refamiliarization with the qualities required for competition, with intensity and duration progressing gradually. Accordingly, the shorter competition format performed after lockdown enabled athletes to maintain a tolerable physiological demand, preventing an excessive cardiovascular stress through the one-day Eventing.
Commented [A13]: Rephrase a little – currently this is hard to follow and understand on the first reading.
Between the three Eventing discipline, DR test resulted the most affected from lockdown. In fact, anxiety and RPE resulted significantly increased post lockdown in the DR test only; and unexpectedly they remained unchanged in the SJ and XC tests. The latter results could be explained by the significantly shorter duration of competition in the post SJ and XC tests, while competition duration was unmodified in the DR post-test.
Commented [A7]: In Italy?
In Italy and France
Commented [A8]: On n horses?
Yes, I added it in text now

Reviewer 2 Report
This is a very interesting study and I have only a couple of small comments for the authors to consider.
Can you clarify what the situation was for the horses during lockdown? As noted, equestrian sport is about human-animal interaction, and if the horses were not trained for 8 weeks they would have lost a significant amount of fitness, suppleness etc. that would not have been fully regained 6 weeks back into training. This would surely have impacted performance.
Related to this, it is interesting that it was the dressage that suffered most, as this is arguably the stage at which the close communication between horse and rider is most significant. If the horses had not been ridden for the 8 weeks - or at least not trained properly in the dressage moves - it is possible that they had become less attuned to the riders. Given that equestrian sport is about this partnership, the role of the horse (as well as the rider and the interaction between the two) is key to performance. So it is not just about rider stress or fatigue, as measured here, but also potentially about horse fitness and suppleness, which may have reduced with lockdown, and horse stress of being back in a competition environment after months away. I realise this was not the focus of this study but some reflection on this is important as it is not just the rider who influences performance.
Personally I find the phrase 'horse-back riding' a bit awkward and suggest the simpler 'horse-riding' reads better.
Author Response
I am really glad you appreciated our work, I hope I fulfilled all you request.
comment: Can you clarify what the situation was for the horses during lockdown? As noted, equestrian sport is about human-animal interaction, and if the horses were not trained for 8 weeks they would have lost a significant amount of fitness, suppleness etc. that would not have been fully regained 6 weeks back into training. This would surely have impacted performance.
replay: During the 8 weeks lockdown most horses could only undergo thirty to forty minutes a day of non-mounted round pen exercises at the three gates. Even worst, they were led by a trainer that usually was not their owner or their usual rider. It can be easily presumed that an insufficient or inappropriate horse training and a premature participation in competition could lead to reduced standards in horse’s welfare.
comment: Related to this, it is interesting that it was the dressage that suffered most, as this is arguably the stage at which the close communication between horse and rider is most significant. If the horses had not been ridden for the 8 weeks - or at least not trained properly in the dressage moves - it is possible that they had become less attuned to the riders. Given that equestrian sport is about this partnership, the role of the horse (as well as the rider and the interaction between the two) is key to performance. So it is not just about rider stress or fatigue, as measured here, but also potentially about horse fitness and suppleness, which may have reduced with lockdown, and horse stress of being back in a competition environment after months away. I realise this was not the focus of this study but some reflection on this is important as it is not just the rider who influences performance.
replay: Moreover, a study conducted on horses participating in elite show jumping and dressage competitions, reported a rather high incidence of conflict behaviour indicative of discomfort, confusion, and resistance or hyperactivity. Conflict behaviours increased in association with increasing levels of complication or difficulty of the tasks encountered, maybe indicative of some shortcoming in the preparation of the equine athletes including overtasking or some other measures of internal conflict or stress being experienced by the horse while being ridden in competitions [26]. Competing with physically or psychologically unprepared horses could lead to mental and/or physical damage to the animal, therefore return to competitions should be carefully planned to ensure, not only successful performances, but mostly horses’ welfare.
26. Górecka-Bruzda, A., KosiÅ„ska, I., Jaworski, Z., Jezierski, T., & Murphy, J. (2015). Conflict behavior in elite show jumping and dressage horses. Journal of Veterinary Behavior, 10(2), 137-146.
comment: Personally I find the phrase 'horse-back riding' a bit awkward and suggest the simpler 'horse-riding' reads better.
replay: changed horse-back riding in of horse-riders along the text

Reviewer 3 Report
Thank you for the opportunity to review the manuscript entitle “Horse-back riding competitions pre and post COVID 19: Effect of anxiety, sRPE, and HR on performance in Eventing” for the International Journal of Environmental Research and Public Health. This manuscript shows unpublished data regarding the impact of training restrictions due to COVID-19 sanitary emergency on the physical and emotional strain of horse-back eventing competitions before and after lockdown. This was an interesting manuscript, and I enjoyed reviewing it. The experimental design and resulting data are suitable to explore an important topic. Overall, it was thought-provoking and enjoyable to read. I only have just a little suggestion to improve the manuscript:
Materials and Methods: authors should extrapolate the parts concerning “statistical analysis” and create a new paragraph.
Discussion: The authors should briefly discuss the role played by “stress” as a factor that in turn influences performance activity in horse-back riding. This could give a greater impact on your article.
Author Response
I am really glad you appreciated our work, I hope I fulfilled all you request.
comment: Materials and Methods: authors should extrapolate the parts concerning “statistical analysis” and create a new paragraph.
replay: as you suggested we created a separate paragraph for statistical analysis
comment: Discussion: The authors should briefly discuss the role played by “stress” as a factor that in turn influences performance activity in horse-back riding. This could give a greater impact on your article.
replay:
Horse riding relies on subtle interactions between the horse and the athlete, performance not depending on human ability and skill only, but equally on that of the equine partner and most importantly on the quality of the interaction between the horse and the rider [4, 25].
To be able to perform ‘‘as one,’’ the aids given by riders should be as subtle and refined as possible, demanding considerable levels of fine motor control and accuracy. Hence, riders’ low levels of arousal are probably advantageous. Increases in muscular tension, respiratory rate, and heart rate are all likely to affect the rider’s ability: first, to transmit an aid, and second, to release the aid again as quickly as possible afterward. Equestrian sports could be anxiety-inducing, especially in times of increased mental and physiological stress such as the return to competitions after a forced interruption of training. Riders may tend to ‘‘hold on’’ too tightly to the reins, or block the horse with their seat, as higher muscular tension would prevent them from relaxing the relevant muscle groups [26]. Physical tension resulting from increased levels of anxiety is detected by the horse, who, in turn, reacts with physiological symptoms associated with anxiety and fear. Any signal indicating anxiety from the rider, such as increased muscle tonus, respiratory or heart rates, may alert the horse, making it fear imminent danger, which is likely to make it react with a flight response, usually leading to faults or loss of marks [4].
Riders mental skills training aimed at improving emotional composure and physical relaxation is likely to also have a beneficial effect on the horse. A study conducted on horses participating in elite show jumping and dressage competitions, reported a rather high incidence of conflict behaviour indicative of discomfort, confusion, and resistance or hyperactivity.
4. Wolframm, I. A., & Micklewright, D. (2010a). Effects of trait anxiety and direction of pre-competitive arousal on performance in the equestrian disciplines of dressage, showjumping and eventing. Comparative Exercise Physiology, 7(4), 185-191. https://doi:10.1017/S1755254011000080
25. McGreevy P. D. (2007). The advent of equitation science. Veterinary journal (London, England: 1997), 174(3), 492–500. https://doi.org/10.1016/j.tvjl.2006.09.008
26. von Borstel, U. U., Duncan, I. J., Lundin, M. C., & Keeling, L. J. (2010). Fear reactions in trained and untrained horses from dressage and show-jumping breeding lines. Applied Animal Behaviour Science, 125(3-4), 124-131.

Reviewer 4 Report
1. Overall I think the paper is interesting and has some good features. These include the multi-method assessment, combining psychological and physiological. 2. The topic seems relatively novel and, of course, topical (though a little 'niche'). 3. The English needs a further edit. 4. A stronger justification for the methods is needed. Why RPE and heart rate, and why the anxiety measure? 5. Strengths and limitations need to be discussed more.
Author Response
I am really glad you appreciated our work, I hope we fulfilled all you request.
I replayed to all comments on the attached file, here only longer replay are reported:
comment: you shown limitations of RPE, but then use it. You need to provide a good justification for its use. More to the point, the issue is about the importance of perceived exertion - why it is important here? Then the measurement issue can be addressed
replay: A study aimed to estimate the psychological impact of COVID-19 and related restrictive measures on the Italian general population during lockdown reported that 52.4% of participants experienced poor sleep and anxiety. Authors also expected mental health symptoms to persist beyond this critical situation [7]. It is perhaps not surprising then, that acute stress disorders, anxiety, irritability, poor concentration and indecisiveness, deteriorating work performance, post-traumatic stress disorders, high psychological distress, depressive symptoms and insomnia are reported as consequences of quarantine [8]. Bompa & Buzzichelli [9] suggest that an abrupt cessation of training by highly trained athletes creates a phenomenon known as detraining syndrome, characterized by insomnia, anxiety, depression, alterations to cardiovascular function, and loss of appetite. Mental fatigue has been demonstrated to negatively influence performance [10] possibly due to an alteration in the perception of effort when individuals are mentally tired [11]. Mental fatigue represents sensations of tiredness experienced during or after prolonged periods of cognitive activity that often influence decision making, attention, motivation, and the voluntary willingness to resist fatigue. Therefore, to ensure that athletes adhere to the program, training load should be individualised and adapted to the possible sensation of tiredness induced by the lockdown, modifying the ratio between prefrontal activation and Rating of Perceived Exertion (RPE) [12]. Consistently, recent studies [13] have suggested that the brain regulates exercise performance based on afferent information from peripheral physiologic receptors that give rise to sensations (e.g. nausea and thirst among others). Indeed, not only, physiological, and neural determinants explain the variation of RPE, but also personality factors (extraversion, neuroticism, depression, and anxiety) were shown to affect RPE [14]. Therefore, application of the RPE method can help to understand the multifaced influence of lockdown on athlete’s fatigue and stress and to carefully manage the athletes back to full training and competition.
7. Gualano, M. R., Lo Moro, G., Voglino, G., Bert, F., & Siliquini, R. (2020). Effects of Covid-19 Lockdown on Mental Health and Sleep Disturbances in Italy. International journal of environmental research and public health, 17(13), 4779. https://doi.org/10.3390/ijerph17134779
8. Brooks, S. K., Webster, R. K., Smith, L. E., Woodland, L., Wessely, S., Greenberg, N., & Rubin, G. J. (2020). The psychological impact of quarantine and how to reduce it: rapid review of the evidence. Lancet (London, England), 395(10227), 912–920. https://doi.org/10.1016/S0140-6736(20)30460-8
9. Bompa, T. O., & Buzzichelli, C. (2018). Periodization-: theory and methodology of training. Human kinetics.
10. Brownsberger, J., Edwards, A., Crowther, R., & Cottrell, D. (2013). Impact of mental fatigue on self-paced exercise. International Journal of Sports Medicine, 34(12), 1029‑1036. https://doi.org/10.1055/s- 0033-1343402
11. Marcora, S. M., Staiano, W., & Manning, V. (2009). Mental fatigue impairs physical performance in humans. Journal of Applied Physiology, 106(3), 857‑864. https://doi.org/10.1152/japplphysiol.91324.2008
12. Borg, G. A. (1982). Psychophysical bases of perceived exertion. Medicine and Science in Sports and Exercise, 14(5), 377‑381.
13. Kilpatrick, M., Kraemer, R., Bartholomew, J., Acevedo, E., & Jarreau, D. (2007). Affective responses to exercise are dependent on intensity rather than total work. Medicine and science in sports and exercise, 39(8), 1417. https://doi: 10.1249/mss.0b013e31806ad73c. PMID: 17762376.
14. Haddad, M., Stylianides, G., Djaoui, L., Dellal, A., & Chamari, K. (2017). Session-RPE Method for Training Load Monitoring: Validity, Ecological Usefulness, and Influencing Factors. Frontiers in neuroscience, 11, 612. https://doi.org/10.3389/fnins.2017.00612
comment: why have you chosen this instrument? It has 3 subscales and these need to be justified.
Replay: The CSAI-2 was used to measure cognitive anxiety, somatic anxiety and self-confidence among the participants. It has 27 items with nine items in each of the three subscales. Possible scores for each sub-scale ranged from 9 to 36. Respondents rate on a 4-point scale the extent of their agreement using anchors of 1: Not at all and 4: Very much. Higher scores on the subscales somatic and cognitive anxiety reflect higher levels of anxiety, a higher score on the self –confidence subscale indicate higher levels of self-confidence. The CSAI-2 measures temporal states, rather than stable personality traits and has good relationship with equestrian performance. In general, anxiety is made up of cognitive and somatic components. Cognitive anxiety encompasses worries or concerns about potential failure and the adequacy of one’s performance as well as disrupted attention and negative expectations. The somatic aspects of anxiety are seen as comprising the autonomic arousal with its physiological responses such as sweating and increased heart rate. Cognitive anxiety is by definition indicative of negative expectations and focused on negative thoughts in the form of worry. Anxiety, as an emotional response, can significantly influence successful return to sport. In addition, anxieties related to the inability and/or uncertainty to return to previous level of performance and lack of athletic appearance have been found to influence a successful return to sport process. Moreover, lack of athletic identity, feelings of isolation, and pressures to return to sport when the athletes themselves do not feel ready to return are also typical emotional responses during the return to sport phase, and they are likely to increase anxiety if not addressed. Moreover, athletes’ affective responses are proven to be related to perception of exercise intensity, being higher intensities associated with less favorable affective responses.
Comment: please add a section on study limitations and strengths
Replay: This study reported some particular aspects, strengths and practical applications. It is the first to quantify the impact of horse-riders training restrictions on physical and emotional strain of Eventing competitions after 8 weeks of lockdown due to the COVID-19 emergency. One strength is the use of objective measures of sports performance, anxiety, exertion, and effort. Moreover, RPE appears to offer a practical method of monitoring athletes work load during training and competition, but it could also be of use for home-confined training during any future sport activities restrictions. It is important to mention that RPE does not concern athletes only but can also provide a needed guidance for the general population fitness maintenance. Lastly, emotional stress in Dressage and work load dependent from course length in Cross-Country should be carefully managed by equestrian Eventing stakeholders supervising athletes, during this unprecedented period of restriction, and when planning training and competitions after a period of lockdown.
Several limitations need to be acknowledged. First, the present study is limited to young female high-level athletes, therefore results could not be applicable to male or beginners horse-riders. Nonetheless, the use of data from studies on elite level competitions will allow novice rider’s coaches to imitate high level riders for appropriate training prescription and profitable competition strategy. Second, the data are correlational, thus conclusions concerning causal relations cannot be drawn. Third, the scales used to measure somatic and cognitive aspects and self-confidence of sport competition trait anxiety were not intended to differentiate between these three aspects of anxiety.
Future research is required to better understand the impact of several weeks of training restrictions, due to any emergency state, on physical and emotional strain of horse-riding Eventing competitions in different gender population and competition levels.
